# Brief communication: Rare ambient saturation during drifting snow occurrences at a coastal location of East Antarctica

Charles Amory and Christoph Kittel

Department of Geography, University of Liège, Liège, Belgium

**Correspondence:** C. Amory (charles.amory@uliege.be)

**Abstract.** Sublimation of snow particles during transport has been recognised as an important ablation process on the Antarctic ice sheet. The resulting increase in moisture content and cooling of the ambient air are thermodynamic negative feedbacks that both contribute to increase the relative humidity of the air, inhibiting further sublimation when saturation is reached. This self-limiting effect and the associated development of saturated near-surface air layers in drifting snow conditions have been
mainly described through modelling studies and few field observations. A set of meteorological data including drifting snow mass fluxes and vertical profiles of relative humidity collected at site D17 in coastal Adelie Land (East Antarctica) during year 2013 is used to study the relationship between saturation of the near-surface atmosphere and the occurrence of drifting snow in a katabatic wind region among the most prone to snow transport by wind. Atmospheric moistening by the sublimation of the windborne snow particles generally results in a strong increase in relative humidity with the magnitude of drifting snow and a
decrease of its vertical gradient, suggesting that windborne-snow sublimation can be an important contributor to the local near-surface moisture budget. Despite a high incidence of drifting snow at the measurement location (60.1 % of the time), saturation, when attained, is however most often limited to a thin air layer below 1 meter above ground. The development of a near-surface saturated air layer up to the highest measurement level of 5.5 m is observed in only 8.2 % of the drifting snow occurrences or 6.3 % of the time and mainly occurs in strong wind speed and drift conditions. This relatively rare occurrence of ambient
saturation is explained by the likely existence of moisture-removal mechanisms inherent to the katabatic and turbulent nature of the boundary-layer flow that weaken the negative feedback of windborne-snow sublimation. Such mechanisms, potentially quite active in katabatic-generated windborne-snow layers all over Antarctica may be very important in understanding the surface mass and atmospheric moisture budgets of the ice sheet by enhancing windborne-snow sublimation.

## 1 Introduction

Drifting and blowing snow, respectively defined as an ensemble of snow particles raised from the ground at a height below and above 2 meters, occur frequently over windswept areas of Antarctica. Continent-wide modelling studies suggest that erosion through divergence of drifting and blowing snow transport and concurrent sublimation of particles during transport currently represent important ablation processes on the ice sheet (e.g., van Wessem et al., 2018; Agosta et al., 2019). When the effective shear stress exerted by the flow on the snow surface exceeds the threshold value for erosion, particles become mobile and

periodically bounce on the surface in a motion mechanism referred to as saltation. In even stronger winds, saltating particles are entrained from the top of the saltation layer by turbulent eddies and enter into suspension without contact with the surface.

The particles transported through saltation and suspension during drifting and blowing snow occurrences interact with the ambient air and influence the thermodynamic structure of the low-level atmosphere. Mass loss experienced by windborne snow particles through sublimation releases water vapour into and removes heat from the surrounding air. Both processes contribute
to an increase in relative humidity of the air in an inherently self-limiting fashion (Déry et al., 1998; Mann et al., 2000; Bintanja, 2000), eventually inhibiting further sublimation when saturation is reached. Since the mass concentration of windborne snow particles decreases rapidly with height above the snow surface, windborne-snow sublimation leads to the development of a downward sensible heat flux together with an upward latent heat flux that can balance each other in strong wind conditions (Bintanja, 2001a). Moisture exchange also occurs between the snow-covered surface and the atmosphere through surface
sublimation but at much lower rates than windborne-snow sublimation because of the greater exposed surface area and the continuous ventilation of snow particles in the air (Schmidt, 1982).

The thermodynamic effects resulting from interactions of windborne snow particles with the atmosphere hinder accurate determination of sublimation rates from classical physical frameworks and automatic weather station data. The usual Monin-Obukhov similarity theory and related bulk relationships cannot be considered valid in transport conditions since the atmo-
spheric surface layer contains a moisture source and a heat sink (Bintanja, 2001b), implying that the requirement of vertical constancy in turbulent fluxes is not met. In addition, the well-known profile method commonly employed to compute turbulent heat fluxes can become hardly applicable in drifting snow because vertical moisture and temperature gradients are weakened by windborne-snow sublimation and turbulent mixing. As the observed gradients reduce, instrumental inaccuracy becomes important compared to gradients and induces comparatively large flux uncertainties which thus strongly amplify with wind
speed (Barral et al., 2014).

The rate of windborne-snow sublimation both affects and depends on the mass concentration of windborne snow particles. It is also interdependent with the temperature and relative humidity gradients across the transport layer, since the water vapour removed from each particle must be compensated by a heat transfer to the particle. In the absence of direct turbulence measurements and despite the instrumental and calculation limitations mentioned above, this suggests that near-surface meteorological
profiles can however provide information on the effect of snow transport on the moisture budget of the lowest atmospheric layers.

The development of near-surface saturated atmospheric layers through the negative feedback of windborne-snow sublimation has been mainly described from modelling approaches (Déry et al., 1998; Bintanja, 2001b) and few field measurements of individual blowing and drifting snow events (Mann et al., 2000; Bintanja, 2001a). The relative frequency of such situations in a
natural environment is however almost undocumented. This study investigates the relationship between low-level atmospheric saturation and the occurrence of drifting snow based on the analysis of detailed meteorological data collected during year 2013 in Adelie Land, a katabatic wind region of coastal East Antarctica among the most prone to snow transport by wind.

**Table 1.** Characteristics of the sensors used at the profile station D17.

| Sensor | Type | Range | Accuracy |
|---|---|---|---|
| Wind speed | Vector A100LK* | 0.2–60 m s$^{-1}$ | 0.1 m s$^{-1}$ |
| Air temperature | Vaisala HMP45A | -39.2–60 °C | 0.4 °C at -20 °C |
| Air relative humidity | Vaisala HMP45A | 0–100 % | 2 % (RH < 90 %) |
| | | | 3 % (RH > 90 %) |
| Snow height | SR50A* | 0.5–10 m | 0.01 m |
| Snow mass flux | second-generation FlowCapt™** | - | - |

\* Manufacturer Campbell Scientific

\*\* Manufacturer IAV Engineering

## 2 Field area and instrumentation

The detailed meteorological profiles and drifting snow measurements presented in this paper were acquired in the framework of an extensive drifting snow observation campaign that was run on the marginal slopes of Adelie Land in January 2010 (Trouvilliez et al., 2014). The campaign involved three unmanned weather stations (namely D3, D17 and D47) placed at different locations along a 100-km long transect to quantify spatial variations in the area. In this study, only the data collected at site D17 (66.7°S, 139.9°E; 450 m above sea level) is used because vertical profiles have been measured only at this location (see Amory (2019) for a detailed account of the measurements and local meteorological conditions).

D17 is situated in an accumulation zone 10 km inland from the coastline, near the downstream end of a sloping ice field. With frequent, strong and persistent katabatic flows originating almost exclusively from south-easterly directions owing to topographic channeling, D17 benefits from a long unobstructed snow-covered fetch of several hundreds of kilometers. This results in the regular occurrence of non-intermittent, well-developed drifting and blowing snow events (Amory, 2019). Strong turbulent mixing due to frequent katabatic flows induces the nearly-constant presence of a quasi-isothermal layer ensuring that vertical changes in relative humidity during transport are mainly governed by particle-air moisture exchanges along the profile.

Wind speed, temperature, and relative humidity were measured at six levels logarithmically spaced (nominal heights 0.8, 1.3, 2, 2.8, 3.9 and 5.5 m above the surface), while changes in elevation due to accumulation and ablation were assessed with a sonic height ranger. The thermo-hygrometers were housed in naturally-ventilated radiation shields. Information on the occurrence and magnitude of drifting snow was retrieved from a second-generation acoustic FlowCapt™ device that was set up vertically and close to the ground to enable detection of the initiation of drifting snow events. The sensor consists of a 1 m long tube containing electroacoustic transducers that convert the acoustic vibration caused by the windborne snow particles colliding with the tube into a snow mass flux (vertically) integrated over the exposed length of the tube. Note that blowing snow may also occur in snow-transport occurrences necessarily identified as drifting snow since in this configuration detection of snow transport is limited to the first metre above ground. To ensure that significant events are detected and electronic or turbulence noise is removed, drifting snow has been considered to occur when the half-hourly mean of the snow mass flux exceeds a

confidence threshold of $10^{-3}$ kg m$^{-2}$ s$^{-1}$ (Amory et al., 2017). The types and specificities of the instruments installed at D17 are summarised in Table 1. Sampling was done every 15 s and the 30-min statistics were stored on a Campbell CR3000 datalogger.

The meteorological profiles used in this study have been collected continuously from January $1^{st}$ until December $30^{th}$ 2013. Although data at D17 is available for a longer period of time (Amory, 2019), this specific year was selected because the thermo-hygrometers remained functional at all levels throughout the year and almost all of the yearly accumulation as recovered by the height ranger occurred in February (Amory et al., 2017), guaranteeing that the measurement heights have undergone relatively little changes during the rest of the year. This is important for consistent time statistics of relative humidity since (i) the proximity with the snow surface, which acts as a moisture source through sublimation influences the vapour pressure of the air and (ii) the additional moisture loading and atmospheric cooling through windborne-snow sublimation at a given elevation above the snow surface partly depends on the snow mass concentration which is a strongly decreasing function of height.

The thermo-hygrometers are factory calibrated to provide relative humidity with respect to liquid water rather than to ice for the whole range of measured temperatures. Goff and Gratch (1945) formulae were then used to convert the raw sensor value into relative humidity with respect to ice (RH) for below-freezing temperatures, using the sensor temperature reports in the conversion. Converted values in excess of 100 % (Fig. S1) were attributed to the limitations of both the instruments and the conversion method and were thus capped to 100 %. Although supersaturations have been reported in Antarctica for clean, cold atmosphere devoid of condensation nuclei (e.g., Genthon et al., 2013), they are not likely to be sustained at D17 because of the relatively high temperatures and windborne snow particles providing a large number of condensation nuclei (Barral et al., 2014).

Inspection at the dataset revealed a significant proportion of occurrences (7 %) for which the lowest humidity sensor reports raw values (i.e., with respect to water) above 100 %. Such values are most likely caused by riming on the probe and/or snow occasionally trapped in the radiation shield due to its proximity with the surface where drift conditions are the most intense. After removal of these spurious data, 16,289 six-level profiles were available for analysis.

## 3 Results

The evolution of the 2-m wind speed ($U_2$), snow mass flux integrated from 0 to 1 m above the surface ($\eta_1$) and relative humidity profiles with time during a drifting snow event recorded at D17 in May 2013 demonstrates that the near-surface air can readily become saturated in drift conditions (Fig. 1). Before the event begins, only a thin layer near the snow surface is saturated as a result of either surface sublimation or a decrease in air temperature. Then, when $U_2$ rises above 10 m s$^{-1}$ triggering erosion with drifting snow fluxes rapidly exceeding $2\ 10^{-1}$ kg m$^{-2}$ s$^{-1}$, saturation (RH = 100 %) is observed up to the highest measurement level in less than 3 hours. When drifting snow begins to weaken on JJ 123.25, the saturated air layer progressively thins down to the lowest measurement level during the 12 hours that follow the cessation of the event. The match in timing between the rise in the drifting snow flux and in RH up to 100 % at all measurements levels strongly suggests that windborne-snow sublimation is responsible for the development of the saturated air layer. The absence of precipitation simulated by the regional climate

model MAR (Agosta et al., 2019) for the fully continental grid point including D17 over this period provides additional support

to this interpretation. In this specific example a timescale of less than 3 hours is necessary for a saturated air layer to grow up to 5.5 m but inspection at the database reveals that for some events ambient saturation can even occur simultaneously to the initiation of drifting snow. Mann et al. (2000) reported similar rapid time delays (1 hour) to reach saturation up to 11 m above the ground during blowing snow episodes at Halley station.

     The structure illustrated in Fig.1 is however repeated along only 8.2 % of overall drifting snow occurrences, indicating

that saturation, when attained, is most often restricted to the first meter above the surface (Fig. 2). On the other hand, a large majority (82 %) of the profiles showing saturation at each measurement level is associated with the occurrence of drifting snow, the rest of them being most likely linked to the prolonged presence of a saturated environment during calm conditions after the cessation of drifting snow events, to atmospheric cooling and/or to maritime air intrusions during cyclonic disturbances (e.g., Gallée, 1996; Kittel et al., 2018). Considering that surface sublimation is less effective at raising the moisture content

of the near-surface air than windborne-snow sublimation because of enhanced particle ventilation in drift conditions, Fig. 2 also suggests that, due to the high incidence of drifting snow at D17 (60.1 % of the time), the occurrence of saturation is predominantly caused by windborne-snow sublimation.

     Even though the development of a saturated air layer extending up to the uppermost measurement level is relatively rare, the occurrence of drifting snow and high relative humidity values are intimately related (Fig. 3). Binned average profiles of RH

are shown for three classes of drift conditions of increasing magnitude, each class being sorted according to three classes of increasing wind speed. When drifting snow does not occur, a general decrease of RH with increasing wind speed is observed at all levels. This results from turbulent mixing and adiabatic warming of near-surface air with the increase in pressure as katabatic flows reach the coast (Gosink, 1989). The saturation vapour pressure of air (with respect to ice) increases strongly with temperature. As the air warms, its saturation vapour pressure increases without substantial changes in moisture content

due to the absence of windborne-snow sublimation, and hence its relative humidity decreases. Only surface sublimation, and possibly residual temperature gradients in that case contribute to the vertical moisture gradient and lead RH to increase when approaching the surface. Contrastingly, in drift conditions a reversal is observed; windborne snow particles sublimate and lead to an increase in RH with wind speed all along the average profile, offsetting the katabatic drying effect. This increase is greatest when drifting snow is most pronounced, and the vertical gradient reduces as $U_2$ becomes stronger and induces more

efficient ventilation and turbulent mixing. The residual gradient is then due to surface sublimation and/or vertical gradients of snow mass concentration and thus of windborne-snow sublimation. For the strongest winds speeds ($U_2 \geq 20 \text{ m s}^{-1}$) associated with strong drift conditions ($\eta_1 \geq 2 \; 10^{-1} \text{ kg m}^{-2} \text{ s}^{-1}$), the profiles display nearly saturated conditions up to the highest measurement level. This demonstrates that windborne-snow sublimation can be a significant contributor to the near-surface moisture budget at D17, whose importance increases with wind speed.

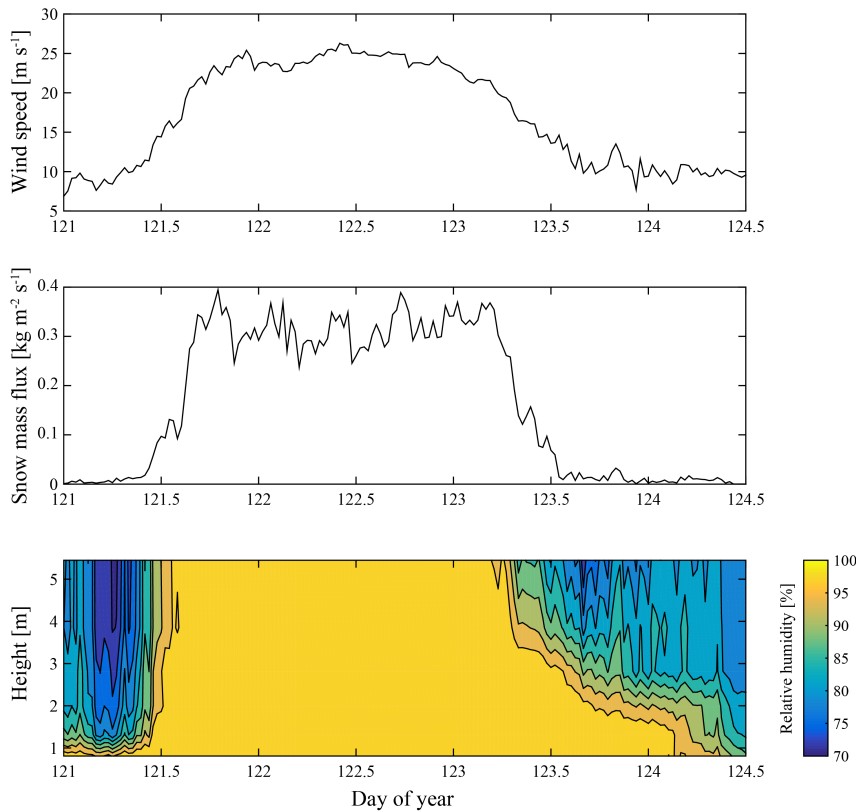

**Figure 1.** Timeseries of 2-m wind speed (upper panel), snow mass flux (middle panel) and contours of relative humidity with respect to saturation over ice (lower panel) showing the development of a saturated air layer during a drifting snow event in May 2013.

## 4  Discussions

Examination of the RH profiles revealed that the air is saturated with respect to ice over the entire measurement range in only 8.2 % of the drifting snow occurrences, assuming saturation is reached when RH = 100 %. Because of instrumental inaccuracy, saturation could however occur at measured RH values below 100 %. The sensitivity of the frequency of saturation can be investigated by decreasing the threshold at which saturation is considered to occur. Accounting for an uncertainty of 3 % in the absolute value of RH as stated by the instrument manufacturer (Table 1), the occurrence of a saturated air layer along the whole measurement range rises from 8.2 % to 18 % of the drifting snow occurrences if the threshold for saturation is lowered from 100 % to 97 %. This twofold increase in the frequency of saturation, globally observed for the upper 4 levels (the lowest 2 levels below 1 m are less significantly affected because of initially high frequency values – Fig. S2), still accounts for a reduced proportion of the overall drifting snow occurrences and confirms that saturation predominantly occurs within the first meter above the snow surface and remains rather infrequent compared to the regular incidence of drift conditions.

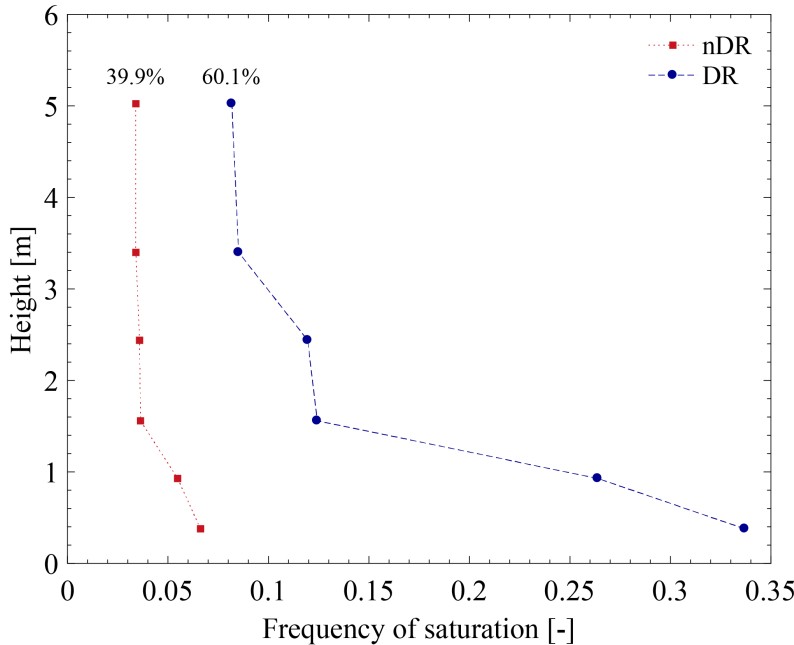

**Figure 2.** Frequency of saturation (RH = 100 %) at each measurement level showing that the saturated air layer that develops in drifting snow rarely exceeds 2 m in height. Profiles collected during non-drift (nDR: $\eta_1 < 10^{-3}$ kg m$^{-2}$ s$^{-1}$) and drift (DR: $\eta_1 \geq 10^{-3}$ kg m$^{-2}$ s$^{-1}$) conditions are treated separately, and their relative proportions are reported at the top of the corresponding curves. Note that saturation frequency values are expressed relative to the total number of profiles for non-drift or drift conditions. Yearly average instrument heights are used..

In previous efforts dedicated to the modelling of windborne-snow sublimation, the saltation layer has been regarded as saturated at all times and emphasis was primarily placed on the sublimation of suspended snow particles. Recent large-eddy simulations (Huang et al., 2016; Sharma et al., 2018) have shown from reassessments of physical starting assumptions that sublimation within the saltation layer might be of significant importance in the surface and atmospheric water budgets. In
some of these simulations, sublimation within the saltation layer can even exceed sublimation of suspended particles by several orders of magnitude when effective advective transport of moisture can sustain an undersaturated environment in the immediate vicinity of the surface (Huang et al., 2016). Assuming that saturation at the lowest measurement level of  0.4 m above ground (i.e., well above the saltation layer) also implies saturation down to the surface, Fig. 2 would indicate that saturation at saltation heights is observed in at least one third of the drifting snow occurrences, i.e. 4 times more frequently than saturated conditions
at all measurement levels. In these cases, sublimation of saltating particles could indeed be ignored and only sublimation within the suspension layer would likely contribute to the surface-atmosphere moisture exchange. Similarly, limited moisture fluxes from sublimation of saltating snow could be expected in fully developed, deep blowing snow layers in which the contribution of suspension to the total column-integrated mass flux dominates over saltation.

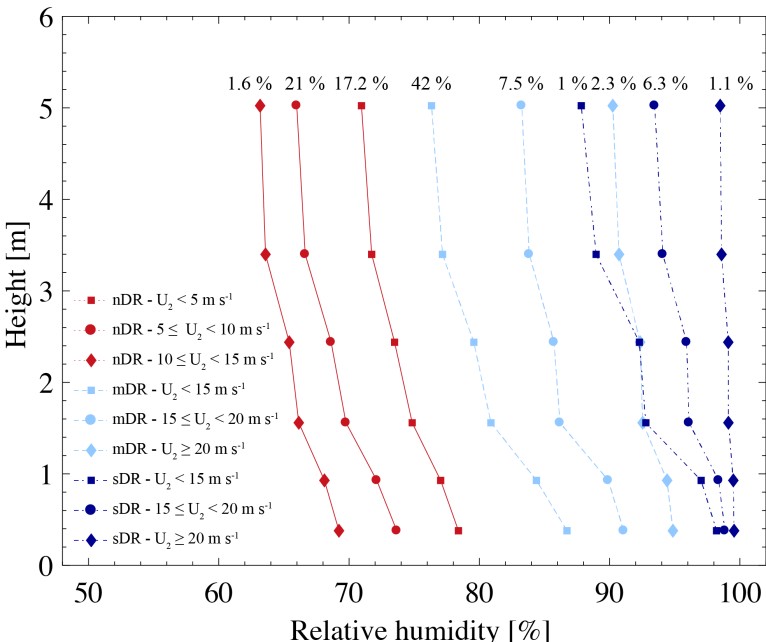

**Figure 3.** 30-min mean profiles of observed relative humidity (with respect to saturation over ice) averaged into bins of increasing 2-m wind speed ($U_2$) and discriminated using the snow mass flux ($\eta_1$) according to non-drift (nDR: $\eta_1 < 10^{-3}$ kg m$^{-2}$ s$^{-1}$), moderate drift (mDR: $10^{-3}$ kg m$^{-2}$ s$^{-1} \leq \eta_1 < 2\ 10^{-1}$ kg m$^{-2}$ s$^{-1}$ and strong drift (sDR: $\eta_1 \geq 2\ 10^{-1}$ kg m$^{-2}$ s$^{-1}$) conditions. The numbers at the top of the average profiles indicate the proportion of observations used to produce each average profile. Yearly average instrument heights are used.

Figure 3 demonstrates that a layer of near-saturated air inevitably develops along the whole profile in the most extreme wind and drift conditions. Similar results obtained from modelling experiments in which a blowing snow layer of 10 m depth is considered, have been used to hypothesise that total windborne-snow sublimation may be limited with strong winds and snow transport because low-level saturation occurs (Bintanja, 2001b). While both results do imply that further moisture release is inhibited in such conditions through the negative feedback of windborne-snow sublimation in a small portion of the near-surface atmosphere, it might also involve that the level of maximum sublimation is lifted to higher elevations when strong winds cause deep blowing snow layers. An important implication could thus be that, despite saturation conditions in the low-level atmosphere, the greatest contribution of windborne-snow sublimation to the total moisture flux might still occur at the highest wind speeds and associated snow transport, as near-surface sublimation is only part of the total atmospheric sublimation.

Although the rare occurrence of ambient saturation indicates that the negative feedback effect of windborne-snow sublimation does not dominate the local moisture budget of the atmospheric boundary layer, this however does not mean that the negative feedback is not at work. Rather it is more likely the result of mechanisms inherent to the katabatic and turbulent nature of the local flow that effectively remove moisture from the transport layer and prevent the formation of a near-surface saturated environment, thereby weakening the negative feedback (Bintanja, 2001b). Examples of such mechanisms are turbulent mixing

within the transport layer, downward entrainment of dry air from above the katabatic layer through wind speed and directional shear, and both horizontal and vertical advection of dry air through adiabatic warming of the descending katabatic flow. Note that windborne snow particles may also contribute to an increase in temperature, and then to a decrease in saturation vapour pressure in the transport layer through additional absorption of longwave radiation (Yang et al., 2014). These moisture-removal mechanisms have been considered as the physical explanation for the persistence of undersaturated air in the upper portion of deep Antarctic blowing snow layers ($> 100$ m) revealed from dropsonde measurements (Palm et al., 2018). The influence of such mechanisms down to lower levels of the transport layer would also explain why the near-surface RH profiles are so infrequently saturated (only in the strongest wind and drift conditions) despite large snow mass fluxes and the attenuating effect due to the increase in RH caused by windborne-snow sublimation. Such mechanisms, potentially quite active in the coastal area of Adelie Land, and more generally in katabatic-generated blowing snow layers all over Antarctica (Bintanja, 2001a; Palm et al., 2018) may be very important in understanding the surface mass and atmospheric moisture budgets of the ice sheet by enhancing windborne-snow sublimation. In contrast, RH profiles collected at Halley over a flat ice shelf, therefore in the absence of a katabatic flow, suggest that the negative feedback of sublimation might locally govern the surface-atmosphere moisture flux (Mann et al., 2000), depending on the dynamical origin of the boundary-layer flow.

## 5  Conclusion

Meteorological profiles and drifting snow mass fluxes collected continuously during year 2013 at site D17 in coastal Adelie Land have been analysed conjointly to characterise the moistening of the near-surface atmosphere during drifting snow occurrences. In snow transport conditions occurring 60.1 % of the time, relative humidity increases and its vertical gradient diminishes with the magnitude of drifting snow, as a result of a major source of moisture from windborne-snow sublimation in the near-surface atmospheric layer. Although saturation is preferentially confined below 1 metre likely involving saturation in the saltation layer, low-level atmospheric moistening by the sublimation of the windborne snow particles can lead to the rapid development of a saturated environment along the whole measurement range that can even coincide with the initiation of drifting snow. However, this is shown to be relatively rare (only 8.2 % of the drifting snow occurrences or 6.3 % of the time if the RH value at which saturation is considered to be attained is taken as 100 %) at the measurement location and mainly occurs in strong drift conditions ($\eta_1 \geq 0.2$ kg m$^{-2}$ s$^{-1}$) associated with high wind speeds ($U_2 \geq 20$ m s$^{-1}$). The low frequency of ambient saturation despite the high incidence of drifting snow is explained by the likely existence of moisture-removal mechanisms that can counterbalance the negative feedback of sublimation by preventing the near-surface air from reaching saturation, raising windborne-snow sublimation as an important contributor to the local surface mass and atmospheric moisture budgets.

The dataset also demonstrates that the occurrence of snow transport does not necessarily involve saturation of the near-surface atmosphere, and conversely the occurrence of a saturated near-surface air layer is not systematically associated with snow transport. But during those events for which the occurrence of drifting snow (i.e., $< 2$ m) and a saturated air layer of several metres is concurrently reported, the height reached by the suspended snow particles, or the depth of the transport layer, likely extends beyond the uppermost measurement level (i.e., $> 5.5$ m). As this possibly induces significant windborne-snow

sublimation in higher atmospheric levels, contrary to previous considerations (Bintanja, 2001b) it is hypothesised that the presence of a low-level saturated air layer would not limit total atmospheric sublimation if the level of maximum sublimation in such conditions is simply moved upwards.

Combined dropsonde and satellite measurements have shown that blowing snow can frequently develop into deep layers of several hundreds of metres above the ice-sheet surface in which RH decreases with height through the depth of the layer (Palm et al., 2018), suggesting the existence of a potentially significant atmospheric mass sink through windborne-snow sublimation. However, satellite detection only captures blowing snow layers 30 m or greater in thickness during clear-sky or optically thin-cloud conditions and is limited by the satellite revisit frequency. Synergistic uses of remote sensing, modelling and observation products would help to assess the proportion of missed events when surveying snow transport from space and study the conditions that lead drifting and shallow (< 30 m) blowing snow events to evolve into deep blowing snow layers, to ultimately improve understanding of the links between snow transport and moisture dynamics and better quantify the influence of windborne-snow sublimation on the surface mass balance of the Antarctic ice sheet

*Data availability.* All data presented and described in this study are freely available by contacting the authors.

*Author contributions.* CA set up the instruments on the field, collected and processed the data, and designed the study. CA and CK wrote the manuscript.

*Competing interests.* The data presented in this study are freely available by contacting the authors.

*Acknowledgements.* This work would not have been possible without the financial and logistical support of the French Polar Institute IPEV (program CALVA-1013). The authors would like to thank all the on-site personnel in Dumont d'Urville and Cap Prud'homme for their precious help in the field, in particular Philippe Dordhain for electronic and technical support, as well as Micheal Lehning and one anynomous reviewer for providing many insightful and constructive comments. C. Amory is a Postdoctoral Researcher from the Fonds de la Recherche Scientifique de Belgique (F.R.S.-FNRS).

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
