# Peer review of "Brief communication: Rare ambient saturation during drifting snow occurrences at a coastal location of East Antarctica"

_The Cryosphere, 2019_

## Referee Comment (RC1) · Anonymous Referee #1 · 6 Oct 2019

Review of

Brief communication: Rare ambient saturation during drifting snow occurrences in coastal East Antarctica

By C. Amory and C. Kittel

General This is a well-written paper with clear figures, and the topic is relevant for The Cryosphere. I do have some reservations about the need for this study to be a separate, brief communication while another study by the same author using the same dataset is presently being considered for publication in the same journal. It must become clearer a) why isolating this aspect of the dataset in a separate publication is

warranted and b) whether the conclusions support the title adopted (see below).

Major comments Why was this paper presented separately from Amory (2019)? Is there a particular reason to do so? An obvious disadvantage is that the reader is now referred to that paper for a detailed description of the observations, which are at the core of this study.

How general are the conclusions? This study only uses a single year of data for a single location. Is this sufficient to support the title of the paper, i.e. that saturation is rare in entire East Antarctica? If maintained, some more effort must go into supporting this claim.

l.79: "...in which a local balance between upward turbulent diffusion, gravitational settling and sublimation of windborne snow particles is likely to be attained." Can you be more specific, do the observations allow to assess in a quantitative way whether such a steady state is attained?

l. 114: "Converted values in excess of 100 % were attributed to the limitations of both the instruments and the conversion method and were thus capped to 100 %." Can you provide some statistics, please? How often did this (RH values > 100%) occur?

Caption Fig. 2: "Frequency of saturation (RH = 100 %)...": given the uncertainty in the RH sensors, saturation could also occur at measured values well below 100%. Have you investigated the sensitivity of your results to this definition of the observed 'saturation' threshold?

l. 123: "Before the event begins, only a thin layer near the snow surface is saturated as a result of surface sublimation.": Alternatively, near-surface air could become saturated not by sublimation but simply by cooling. See also l. 138.

Minor comments l. 46: "The thermodynamic effects of windborne-snow sublimation are physical limitations to accurate determination of sublimation rates from automatic weather station data. ": This is either an awkwardly formulated sentence or it contains

a typo; please reformulate.

l. 52: "…raising instrumental accuracy as a large source of uncertainty which strongly amplifies with wind speed": please explain why instrumental inaccuracy amplifies with wind speed; one could also argue that stability corrections become less important during near-neutrality, enhancing accuracy?

l. 82: "…isothermal layer…": Strictly speaking, a neutral surface layer implies that potential temperature is constant with height, not temperature.

L. 95: emerged -> exposed (?)

---

## Author Comment (AC1) · 8 Oct 2019

We thank the reviewer for its thorough reading of our paper, the very relevant questions and the proposed suggestions. Our responses are reported hereafter.

Response to reviewer #1

General comments This is a well-written paper with clear figures, and the topic is relevant for The Cryosphere. I do have some reservations about the need for this study to be a separate, brief communication while another study by the same author using the same dataset is presently being considered for publication in the same journal. It must

become clearer a) why isolating this aspect of the dataset in a separate publication is warranted and b) whether the conclusions support the title adopted (see below).

Major comments RC1: Why was this paper presented separately from Amory (2019)? Is there a particular reason to do so? An obvious disadvantage is that the reader is now referred to that paper for a detailed description of the observations, which are at the core of this study.

Authors: While Amory (2019) uses single-level meteorological measurements of the full 9-year dataset to focus exclusively on drifting snow frequency and mass transport statistics and discuss applications for model evaluation, in the present paper we use a part (1 year) of this dataset and extend the analysis to 6-level meteorological profiles to discuss a different, independent topic related to the development of saturated air layers. Although the two papers do have some data in common and share a common context of drifting snow features, note that

1) the analysis proposed here relies on specific requirements that are only met during that specific period (availability of the depth sensor, no dysfunction nor burial of any of the 6 thermo-hygrometers, relative constancy in measurement height along the observation period - see the paragraph dedicated to that aspect of the study from L102 to L110), 2) the specific subject of drifting snow atmospheric interactions and sublimation involve theoretical descriptions that does not fit the objective of Amory (2019), and would have thus require additional out-of-the-scope theoretical background, 3) the meteorological profiles are not presented in Amory (2019).

For the above-mentioned reasons, and because we are also deeply convinced that keeping the scientific message of a paper as onefold improves clarity, readability and efficiency and thus prevent the paper from being too long with various scientific messages and disconnected sections, we believe that each of the two studies deserve separate papers. Moreover, the method section of the present paper includes a detailed description of the observations that contains the information (i.e., sensor type, range,

accuracy, measurement principle and measurement height) necessary for the self-sufficiency of the paper, and relies on Amory (2019) only for additional, non-essential information about the observation campaign.

RC1: How general are the conclusions? This study only uses a single year of data for a single location. Is this sufficient to support the title of the paper, i.e. that saturation is rare in entire East Antarctica? If maintained, some more effort must go into supporting this claim. Authors: You are entirely right, Antarctica is wide and diverse. We suggest to change the title to "Rare ambient saturation during drifting snow occurrences at a coastal location of East Antarctica".

RC1: L79: ". . .in which a local balance between upward turbulent diffusion, gravitational settling and sublimation of windborne snow particles is likely to be attained." Can you be more specific, do the observations allow to assess in a quantitative way whether such a steady state is attained? Authors: This indeed may have been extrapolated a bit too far, at least in assuming steady-state drifting snow. Answering this question in a quantitative way could ideally be done through a modelling approach but observations alone hardly allows it. This part has been removed from the sentence.

RC1: L114: "Converted values in excess of 100 % were attributed to the limitations of both the instruments and the conversion method and were thus capped to 100 %." Can you provide some statistics, please? How often did this (RH values > 100%) occur? Authors: RH is the ratio of water vapour pressure to the vapour pressure for saturated air. RH values converted to be given with respect to ice (RH w.r.i.) are necessarily higher than original values given with respect to water (RH w.r.w.), since ice supports a lower saturation vapour pressure than water for a given temperature. A comparison between raw and converted RH values for the measurement level closest to 2 m is given in Fig. R1. The figure shows that converted RH values slightly exceeding 100 % (up to a maximum of 105%) occur regularly and inevitably account for most of the saturation conditions (considering that saturation is reached at RH >=100 %), as a result of the empirical character and inherent limitation of the widely used Goff-Gratch

formulae. As the occurrence of supersaturation is very unlikely at our measurement site (as discussed in the paper from L115 to L118 in the original version), and the conversion of raw sensor records is needed for a matter of rigour, we maintain that Goff-Gratch conversion is a reasonable option and suggest to let the text as it appears in the original version of the manuscript but to add Fig. R1 as Fig. S1 in the supplementary materials.

RC1: Caption Fig. 2: "Frequency of saturation (RH = 100 %). . .": given the uncertainty in the RH sensors, saturation could also occur at measured values well below 100%. Have you investigated the sensitivity of your results to this definition of the observed 'saturation' threshold? Authors: Accounting for an uncertainty of 3 % in the absolute value of RH as stated by the manufacturer (Table 1 of the original version), the occurrences of a saturated air layer along the whole measurement range rise from 9.7 % to 21.3 % of the drifting snow occurrences if the saturation threshold is lowered from 100 % to 97 %. This twofold increase in the frequency of saturated conditions, also observed for the upper 4 levels (the lowest 2 levels below 1 m are less significantly affected because of initially high frequency values), still accounts for a reduced proportion of the overall drifting snow occurrences and confirm that saturation predominantly occurs within the first two meters above the surface and remains rather infrequent compared to the regular incidence of drift conditions.

We have added the following paragraph at the beginning of the discussion section, and Fig. R2 is now proposed as Fig. S2 in the supplementary materials: "Examination of the RH profiles revealed that the air is saturated with respect to ice over the entire measurement range in only 10.8 % of the drifting snow occurrences, assuming saturation is reached when RH = 100 %. Because of instrumental inaccuracy, saturation could however occur at measured RH values below 100 %. The sensitivity of the frequency of saturation can be investigated by decreasing the threshold at which saturation is considered to occur. Accounting for an uncertainty of 3 % in the absolute value of RH as stated by the instrument manufacturer (Table 1), the occurrence of a saturated air

layer along the whole measurement range rises from 10.8 % to 23.2 % of the drifting snow occurrences if the threshold for saturation is lowered from 100 % to 97 %. This twofold increase in the frequency of saturation, globally observed for the upper 4 levels (the lowest 2 levels below 1 m are less significantly affected because of initially high frequency values – Fig. S2), still accounts for a reduced proportion of the overall drifting snow occurrences and confirms that saturation predominantly occurs within the first two meters above the surface and remains rather infrequent compared to the regular incidence of drift conditions.".

The sentence in the conclusion from L214 to L216 in the original version has been completed by mentioning the threshold retained for the computation of the statistics as follows: "[...]. However, this is shown to be relatively rare (only 10.8 % of the drifting snow occurrences or 6.5% of the time if the RH value at which saturation is considered to be attained is taken as 100 %) at the measurement location and mainly occurs in strong drift conditions (> 0.2 kg m-2 s-1) associated with high wind speeds ($U\_2$ > 20 m s-1).".

Moreover, by re-investigating the dataset we have also discovered significant occurrences (7% of the data) for which the lowest humidity sensor reports raw RH (with respect to water) above 100 %, most likely due to riming on the probe and/or snow trapped in the radiation shield caused by its proximity with the surface where drift conditions are the most intense. These observations have been removed from the dataset and the (slightly modified) statistics and figures have been corrected accordingly. The following paragraph has been added to the method section: "Inspection at the dataset revealed a significant proportion of occurrences (7%) for which the lowest humidity sensor reports raw values (i.e., with respect to water) above 100 %. Such values are most likely caused by riming on the probe and/or snow occasionally trapped in the radiation shield due to its proximity with the surface where drift conditions are the most intense. After removal of these data, 16,289 six-level profiles were available for analysis." . RC1: L123: "Before the event begins, only a thin layer near the snow surface is

saturated as a result of surface sublimation.": Alternatively, near-surface air could become saturated not by sublimation but simply by cooling. See also L138. Authors: Yes it is true, a decrease in air temperature can just as likely as surface sublimation cause saturation in the vicinity of the surface. This suggestion has been added in the text as "[...] saturated as a result of surface sublimation and/or a decrease in air temperature". Atmospheric cooling has also been listed as a possible explanation for the persistence of saturated conditions after the cessation of drifting snow events (L138 in the original version).

Minor comments RC1: L46: "The thermodynamic effects of windborne-snow sublimation are physical limitations to accurate determination of sublimation rates from automatic weather station data. ": This is either an awkwardly formulated sentence or it contains a typo; please reformulate. Authors: We tried to resume with this first sentence the content of the underlying paragraph, but surely we did it in a clumsy way. We suggest to change the sentence to "The thermodynamic effects resulting from interactions of drifting snow particles with the atmosphere hinder accurate determination of sublimation rates from classical physical frameworks and automatic weather station data."

RC1: L52: ". . .raising instrumental accuracy as a large source of uncertainty which strongly amplifies with wind speed": please explain why instrumental inaccuracy amplifies with wind speed; one could also argue that stability corrections become less important during near-neutrality, enhancing accuracy? Authors: Turbulent mixing and drifting snow promote a weakening in the observed temperature and moisture gradients, to the extent that measurement accuracy are of the order of the existing gradients (Barral et al. 2014). In addition, increasing wind speeds are generally accompanied with an increase in relative humidity (Fig. 3 in the submitted version), resulting in a strong sensitivity of the profile method to measurement errors, particularly in the case of small gradients in conjunction with strong winds. For further details, we refer to Barral et al. (2014), in particular their section 5.2 and Fig. 12 which show that "the propagated uncertainties are amplified with wind velocity or decreasing temperature gradients". Even if a significant influence of stability corrections in the computation of turbulent fluxes has been demonstrated for stable conditions on the Antarctic plateau (Vignon et al., 2017), a much lower influence can be expected at D17 because the near-surface atmosphere is mostly neutrally stratified throughout the year, and it is more likely that the gradient-induced uncertainty largely offsets the enhanced accuracy due to the decreasing importance of stability corrections with increasing wind speeds. To improve clarity, we suggest to reformulate the sentence L50 to L53 in the original version as "In addition, the well-known profile method commonly employed to compute turbulent heat fluxes can become hardly applicable in drifting snow because vertical moisture and temperature gradients are weakened by windborne-snow sublimation and turbulent mixing. As the observed gradients reduce, instrumental inaccuracy becomes important compared to gradients and induces comparatively large flux uncertainties which thus strongly amplify with wind speed (Barral et al., 2014).".

RC1: L82: ". . .isothermal layer. . .": Strictly speaking, a neutral surface layer implies that potential temperature is constant with height, not temperature. Authors: Absolutely right. This sentence has been changed to "Strong turbulent mixing due to frequent katabatic flows induces the nearly-constant presence of an isothermal layer ensuring that vertical changes in relative humidity during transport are mainly governed by particle-air moisture exchanges along the profile".

RC1: L. 95: emerged -> exposed (?) Authors: Corrected accordingly.

Amory, C.: Drifting snow statistics from multiple-year autonomous measurements in Adelie Land, eastern Antarctica, The Cryosphere Discuss., https://doi.org/10.5194/tc-2019-164, in review, 2019.

Barral, H., Genthon, C., Trouvilliez, A., Brun, C. and Amory, C.: Blowing snow in coastal Adélie Land, Antarctica: three atmospheric-moisture issues, The Cryosphere, 8(5), 1905–1919, doi:10.5194/tc-8-1905-2014, 2014.

Vignon, E., Genthon, C., Barral, H., Amory, C., Picard, G., Gallée, H., Casasanta, G. and Argentini, S.: Momentum and heat-fluxparameterization at Dome C, Antarctica: a sensitivity study, Boundary-Layer Meteorol., 162, 341–367, https://doi.org/10.1007/s10546-016-0192-3, 2016.

[Figure]

**Figure R1.** Timeserie of raw (given with respect to water; w.r.w.) and converted (given with respect to ice; w.r.i.) RH values at 2 m height showing the regular occurrence of RH > 100 % due to limitations of the Goff-Gratch formulae used in the conversion.

**Fig. 1.**

[Figure]

Figure R2. Frequency of saturation for in drift conditions at each measurement level assuming that saturation is reached at 100 % (dashed line) and 97 % (sold line). Yearly average instrument heights are used.

**Fig. 2.**

---

## Referee Comment (RC2) · Michael Lehning (Referee) · 12 Oct 2019

Review "Brief communication: Rare ambient saturation during drifting snow occurrences in coastal East Antarctica" by Charles Amory and Christoph Kittel

Review by M. Lehning

The paper presents an observational study on moisture dynamics at one site in Antarctica with frequent events of strong katabatic winds and associated snow transport. The paper is nicely formulated and timely as the role of sublimation during snow transport is a currently debated problem in meteorology and snow science.

I asked myself whether this rather limited dataset is worth a separate publication but concluded that it helps to shape our understanding of what blowing snow sublimation may look like in these katabatic wind zones. However, I suggest that more complete context and discussion is provided. There have been recent LES simulations on drifting snow sublimation (e.g. Sharma et al., 2018; Huang et al., 2016), which claim that previous efforts in modelling sublimation may have started from wrong assumptions. It would add value and impact to the current paper if the authors could discuss whether their observations are consistent with the new model findings or not.

One major comment I have is on the overall limitation of sublimation in snow transport clouds. The authors revisit the argument that with stronger wind and snow transport, total snow sublimation may be limited because saturation occurs. This argument has been formulated by Bintanja (2001) based on a model study, in which the author considers a model depth of 10 m to look of blowing snow sublimation. I have always been very skeptical about the conclusion of limited sublimation because I expect in these situations the level of maximum sublimation to be simply lifted to higher elevations such that it is not seen in the first 10 m. If I understand the Bintanja model study correctly, then he only sums up sublimation occurring in the lowest 10 m, which is of course only part of the total sublimation if high winds cause deep clouds of blowing snow. The authors are therefore encouraged to either present clear evidence of total sublimation reduction in stronger winds, or not conclude about this aspect.

Minor comments:

In practical meteorological applications, drifting snow is below 2 m and blowing snow above. Snow scientists, however, would rather define drifting snow as saltation and blowing snow as suspended snow.

l. 30 (and first sentence in the abstract): I think that it is not proven yet or generally accepted that snow transport and sublimation is the main ablation process over the entire Antarctic ice sheet.

[Figure]

l. 50: Suggest to replace "verified" by "met".

l. 75 vs. l. 80: You cannot say that D17 is in an accumulation zone and then that you achieve equilibrium horizontal mass flux. This is contradictory.

l. 107: Whether or not the vapor pressure really increases towards the surface (or only most often) depends on the air temperature gradient (and the one in the surface snow to a lesser degree).

l. 161 ff: Mark this as hypothesis/discussion and mention (again) temperature gradients, which are also able to produce moisture gradients.

l. 175: See major comment above on moving the elevation of maximum sublimation upwards in higher winds. This is what I expect to occur.

l. 212: Check wording "preferably".

References:

Bintanja, R.: Modelling snowdrift sublimation and its effect on the moisture budget of the atmospheric boundary layer, Tellus A, 53(2), 215–232, doi:10.1034/j.1600-0870.2001.00173.x, 2001.

Sharma, V., Comola, F., and Lehning, M.: On the suitability of the Thorpe-Mason model for Calculating Sublimation of Saltating Snow, The Cryosphere, 12(11). 3499-3509, https://doi.org/10.5194/tc-12-3499-2018, 2018.

Huang, N., X. Dai, and J. Zhang (2016), The impacts of moisture transport on drifting snow sublimation in the saltation layer, Atmospheric Chemistry & Physics, 16(12), 7523-7529.

---

## Author Comment (AC2) · 25 Oct 2019

Dear Michael,

We thank you for your insightful comments that will undoubtedly help to improve the quality of the manuscript. We have adapted the text in several places and included additional discussions to fit your suggestions. Our responses are reported hereafter.

General comments

The paper presents an observational study on moisture dynamics at one site in Antarctica with frequent events of strong katabatic winds and associated snow transport. The

paper is nicely formulated and timely as the role of sublimation during snow transport is a currently debated problem in meteorology and snow science.

Major comments

RC2: I asked myself whether this rather limited dataset is worth a separate publication but concluded that it helps to shape our understanding of what blowing snow sublimation may look like in these katabatic wind zones. However, I suggest that more complete context and discussion is provided. There have been recent LES simulations on drifting snow sublimation (e.g. Sharma et al., 2018; Huang et al., 2016), which claim that previous efforts in modelling sublimation may have started from wrong assumptions. It would add value and impact to the current paper if the authors could discuss whether their observations are consistent with the new model findings or not.

Authors: Thank you for this very relevant suggestion. We have completed the discussion with the following paragraph in which Fig. 2 is used to discuss our results from the perspective of the above-mentioned model findings: "In previous efforts dedicated to the modelling of windborne-snow sublimation, the saltation layer has been regarded as saturated at all times and emphasis was primarily placed on the sublimation of suspended snow particles. Recent large-eddy simulations (e.g., Huang et al., 2016; Sharma et al., 2018) have argued from reassessments of physical starting assumptions that sublimation within the saltation layer might be of significant importance in the surface and atmospheric water budgets. In some of these experiments, sublimation within the saltation layer can even exceed sublimation of suspended particles by several orders of magnitude when effective advective transport of moisture can sustain an undersaturated environment in the immediate vicinity of the surface (Huang et al., 2016). Assuming that saturation at the lowest measurement level of $\sim$0.4 m above ground (i.e., well above the saltation layer) also implies saturation down to the surface, Fig. 2 would indicate that saturation at saltation heights is observed in at least one third of the drifting snow occurrences, i.e. 4 times more frequently than saturated conditions at all measurement levels. In these cases, sublimation of saltating particles could

indeed be ignored and only sublimation within the suspension layer would likely contribute to the surface-atmosphere moisture exchange. Similarly, limited moisture fluxes from sublimation of saltating snow could be expected in fully developed, deep blowing snow layers in which the contribution of suspension to the total column-integrated mass flux dominates over saltation.".

A brief mention of this new element of discussion is also made in the conclusion by completing the former sentence "Although saturation is preferentially confined below 1 metre, low-level atmospheric moistening by the sublimation of [...]" which thus becomes "Although saturation is preferentially confined below 1 metre likely involving saturation in the saltation layer, low-level atmospheric moistening by the sublimation of [...]".

RC2: One major comment I have is on the overall limitation of sublimation in snow transport clouds. The authors revisit the argument that with stronger wind and snow transport, total snow sublimation may be limited because saturation occurs. This argument has been formulated by Bintanja (2001) based on a model study, in which the author considers a model depth of 10 m to look of blowing snow sublimation. I have always been very skeptical about the conclusion of limited sublimation because I expect in these situations the level of maximum sublimation to be simply lifted to higher elevations such that it is not seen in the first 10 m. If I understand the Bintanja model study correctly, then he only sums up sublimation occurring in the lowest 10 m, which is of course only part of the total sublimation if high winds cause deep clouds of blowing snow. The authors are therefore encouraged to either present clear evidence of total sublimation reduction in stronger winds, or not conclude about this aspect.

Authors: We could not agree more with your comment. We have removed the conclusion about the original argumentation on the peak in sublimation for moderate winds in the Results section and added the following paragraph in the Discussion section"Figure 3 demonstrates that a layer of near-saturated air inevitably develops along the whole profile in the most extreme wind and drift conditions. Similar results obtained from

modelling experiments in which a blowing snow layer of 10 m depth is considered, has been used to hypothesise that total windborne-snow sublimation may be limited with strong winds and snow transport because low-level saturation occurs (Bintanja, 2001b). While both results do imply that further moisture release is inhibited in such conditions through the negative feedback of windborne-snow sublimation in a small portion of the near-surface atmosphere, it might also involve that the level of maximum sublimation is lifted to higher elevations when strong winds cause deep blowing snow layers. An important implication could thus be that, despite saturation conditions in the low-level atmosphere, the greatest contribution of windborne-snow sublimation to the total moisture flux might still occur at the highest wind speeds and associated snow transport, as near-surface sublimation is only part of the total atmospheric sublimation.".

As we believe that this reasoning can be of significant relevance for future blowing snow studies, it is also mentioned in the conclusion as: "[. . .] during those events for which the occurrence of drifting snow (i.e., < 2 m) and a saturated air layer of several metres is concurrently reported, the height reached by the suspended snow particles, or the depth of the transport layer, likely extends beyond the uppermost measurement level (i.e., > 5.5 m). As this possibly induces significant windborne-snow sublimation in higher atmospheric levels, contrary to previous considerations (Bintanja, 2001b) it is hypothesized that the presence of a low-level saturated air layer would not limit total atmospheric sublimation if the level of maximum sublimation in such conditions is simply moved upwards.".

Finally we have adapted our last paragraph in the conclusion as: "Combined dropsonde and satellite measurements have shown that blowing snow can frequently develop into deep layers of several hundreds of metres above the ice-sheet surface in which RH decreases with height through the depth of the layer (Palm et al., 2018), suggesting the existence of a potentially significant atmospheric mass sink through windborne-snow sublimation. However, satellite detection only captures blowing snow layers 30 m or

greater in thickness during clear-sky or optically thin-cloud conditions and is limited by the satellite revisit frequency. Synergistic uses of remote sensing, modelling and observation products would help to assess the proportion of missed events when surveying snow transport from space and study the conditions that lead drifting and shallow (< 30 m) blowing snow events to evolve into deep blowing snow layers, to ultimately improve understanding of the links between snow transport and moisture dynamics and better quantify the influence of windborne-snow sublimation on the surface mass balance of the Antarctic ice sheet.".

Minor comments

RC2: In practical meteorological applications, drifting snow is below 2 m and blowing snow above. Snow scientists, however, would rather define drifting snow as saltation and blowing snow as suspended snow.

Authors: We agree that a standard definition for distinguishing between drifting and blowing snow is still lacking in the scientific community. The formulation of an official consensus may have been hindered by the use of analogous terminologies for the description of diverse meteorological conditions, (without systematic detailed explanations of the semantics used) to the extent that it would, maybe, ideally require a conciliation meeting some day. However, the definition using a standard level of 2 m in height as a distinction criteria has been widely employed over the recent years in publications dealing with snow transport in Antarctica (e.g., Leonard et al., 2011; Lenaerts et al., 2012; Gossart et al., 2017; Trouvilliez et al., 2014; Palm et al., 2017, 2018a,b) while we found no clear evidence of a distinction involving analogy between drifting and saltating snow from one hand and blowing and suspended snow on the other hand, even in the most recent publications dedicated to the characterization of specific aeolian processes (e.g., Aksamit and Pomeroy, 2017; Crivelli et al., 2016; Huang et al., 2016; Huang and Wang, 2016; Comola and Lehning, 2017; Comola et al., 2017; Paterna et al., 2016, 2017; Sharma et al., 2018). In addition, and from our point of view, one could argue that this requires the use of different words for referring to the same mechanism

(saltation/drifting and suspension/blowing) which could eventually be a bit confusing in some way. One option could be to mention the existence of the two definitions, and be explicit about the one we choose in the rest of the manuscript. In that case, could you indicate publications in which this definition clearly appears so we can refer to them and then support the proposed definition in the paper?

RC2: L30 (and first sentence in the abstract): I think that it is not proven yet or generally accepted that snow transport and sublimation is the main ablation process over the entire Antarctic ice sheet.

Authors: Yes you're right, this assertion actually relies only on a few modelling studies which have not resulted in a consensual acceptance. We changed the sentence to "Continent-wide modelling studies suggest that erosion through divergence of drifting and blowing snow transport and concurrent sublimation of particles during transport currently represent important ablation processes on the ice sheet.".

RC2: L50: Suggest to replace "verified" by "met".

Authors: Changed accordingly.

RC2: L75 vs. L80: You cannot say that D17 is in an accumulation zone and then that you achieve equilibrium horizontal mass flux. This is contradictory.

Authors: This clumsy part has been removed from the sentence.

RC2: L107: Whether or not the vapor pressure really increases towards the surface (or only most often) depends on the air temperature gradient (and the one in the surface snow to a lesser degree).

Authors: The influence of the air temperature gradient on the water vapour gradient decreases with the air temperature gradient itself. Active katabatic winds in the measurement area provide efficient turbulent mixing in the near-surface atmosphere. Figure R1 shows that the air temperature gradient (computed from the temperature difference between the highest and the lowest measurement level) decreases as wind speed in-

creases, and is in most cases below the actual instrumental accuracy of 0.4 °C at -20 °C (see Table 1 in the original manuscript). We can thus expect the actual temperature gradients in the thin portion of the atmospheric boundary layer covered by the meteorological profiles (the first 6 meters above the surface) to be very small (actually below instrumental accuracy), even more in case of moderate to strong winds so we can reasonably make the assumption of the existence of a nearly-isothermal layer. Figure 3 in the original manuscript evidences however the existence of RH gradients above instrumental accuracy for every range of wind speeds that cannot be entirely explained by possible residual temperature gradients; explanations for this gradients have then been proposed from L107 to L110 (in the original version) and used as arguments to justify the importance of relatively constant measurement heights in the statistical analysis of relative humidity. Nevertheless, we have modified slightly the sentence in the revised version of the manuscript and mention instead the proximity with the surface as a influencing factor for RH as :"This is important for consistent time statistics of relative humidity since (i) the proximity with the snow surface, which acts as a moisture source through sublimation influences the vapour pressure of the air and (ii) the additional moisture loading and atmospheric cooling through windborne-snow sublimation at a given elevation above the snow surface partly depends on the snow mass concentration which is a strongly decreasing function of height.".

RC2: L161 ff: Mark this as hypothesis/discussion and mention (again) temperature gradients, which are also able to produce moisture gradients.

Authors: This paragraph now includes temperature gradients as a possible explanation for the RH gradients as "Only surface sublimation, and possibly residual temperature gradients in that case contribute to the vertical moisture gradient and leads RH to increase when approaching the surface.".

RC2: L175: See major comment above on moving the elevation of maximum sublimation upwards in higher winds. This is what I expect to occur.

Authors: See our response to major comment #2.

RC2: L212: Check wording "preferably".

Authors: We changed to "preferentially".

Aksamit, N. O. and Pomeroy, J. W.: The Effect of Coherent Structures in the Atmospheric Surface Layer on Blowing-Snow Transport, Boundary-Layer Meteorol., doi:10.1007/s10546-017-0318-2, 2017.

Bintanja, R.: Modelling snowdrift sublimation and its effect on the moisture budget of the atmospheric boundary layer, Tellus A, 53(2), 215–232, doi:10.1034/j.1600-0870.2001.00173.x, 2001.

Comola, F. and Lehning, M.: Energy‐ and momentum‐conserving model of splash entrainment in sand and snow saltation, Geophys. Res. Lett., 44(3), 1601–1609, doi:10.1002/2016GL071822, 2017.

Comola, F., Kok, J. F., Gaume, J., Paterna, E. and Lehning, M.: Fragmentation of wind-blown snow crystals, Geophys. Res. Lett., 44(9), 4195–4203, doi:10.1002/2017GL073039, 2017.

Crivelli, P., Paterna, E., Horender, S. and Lehning, M.: Quantifying Particle Numbers and Mass Flux in Drifting Snow, Boundary-Layer Meteorol., 161(3), 519–542, doi:10.1007/s10546-016-0170-9, 2016.

Gossart, A., Souverijns, N., Gorodetskaya, I. V., Lhermitte, S., Lenaerts, J. T. M., Schween, J. H., Mangold, A., Laffineur, Q. and van Lipzig, N. P. M.: Blowing snow detection from ground-based ceilometers: application to East Antarctica, The Cryosphere, 11(6), 2755–2772, doi:10.5194/tc-11-2755-2017, 2017.

Huang, N. and Wang, Z.-S.: The formation of snow streamers in the turbulent atmosphere boundary layer, Aeolian Research, 23, 1–10, doi:10.1016/j.aeolia.2016.09.002, 2016.

Huang, N., Dai, X. and Zhang, J.: The impacts of moisture transport on drifting snow sublimation in the saltation layer, Atmos. Chem. Phys., 16(12), 7523–7529, doi:10.5194/acp-16-7523-2016, 2016.

Lenaerts, J. T. M., van den Broeke, M. R., Déry, S. J., van Meijgaard, E., van de Berg, W. J., Palm, S. P. and Sanz Rodrigo, J.: Modeling drifting snow in Antarctica with a regional climate model: 1. Methods and model evaluation, J. Geophys. Res., 117(D5), n/a-n/a, doi:10.1029/2011JD016145, 2012.

Leonard, K. C., Tremblay, L.-B., Thom, J. E. and MacAyeal, D. R.: Drifting snow threshold measurements near McMurdo station, Antarctica: A sensor comparison study, Cold Regions Science and Technology, 70, 71–80, doi:10.1016/j.coldregions.2011.08.001, 2011.

Palm, S. P., Kayetha, V., Yang, Y. and Pauly, R.: Blowing snow sublimation and transport over Antarctica from 11 years of CALIPSO observations, The Cryosphere, 11(6), 2555–2569, doi:10.5194/tc-11-2555-2017, 2017.

Palm, S. P., Yang, Y., Kayetha, V. and Nicolas, J. P.: Insight into the Thermodynamic Structure of Blowing-Snow Layers in Antarctica from Dropsonde and CALIPSO Measurements, J. Appl. Meteor. Climatol., 57(12), 2733–2748, doi:10.1175/JAMC-D-18-0082.1, 2018a.

Palm, S. P., Kayetha, V. and Yang, Y.: Toward a Satellite-Derived Climatology of Blowing Snow Over Antarctica, J. Geophys. Res. Atmos., 123(18), 10,301-10,313, doi:10.1029/2018JD028632, 2018b.

Paterna, E., Crivelli, P. and Lehning, M.: Decoupling of mass flux and turbulent wind fluctuations in drifting snow, Geophys. Res. Lett., 43(9), 4441–4447, doi:10.1002/2016GL068171, 2016.

Paterna, E., Crivelli, P. and Lehning, M.: Wind tunnel observations of weak and strong snow saltation dynamics, J. Geophys. Res. Earth Surf., 122(9), 1589–1604,

doi:10.1002/2016JF004111, 2017.

Sharma, V., Comola, F. and Lehning, M.: On the suitability of the Thorpe–Mason model for calculating sublimation of saltating snow, The Cryosphere, 12(11), 3499–3509, doi:10.5194/tc-12-3499-2018, 2018.

Trouvilliez, A., Naaim-Bouvet, F., Genthon, C., Piard, L., Favier, V., Bellot, H., Agosta, C., Palerme, C., Amory, C. and Gallée, H.: A novel experimental study of aeolian snow transport in Adelie Land (Antarctica), Cold Regions Science and Technology, 108, 125–138, doi:10.1016/j.coldregions.2014.09.005, 2014.
* * *
[Figure]

**Fig. 1.** Figure R1. Air temperature gradient (computed from the temperature difference be-
tween the highest and the lowest measurement level) as a function of wind speed.